# Relationship between the COVID-19 Pandemic and the Well-Being of Adolescents and Their Parents in Switzerland

**DOI:** 10.3390/ijerph19116789

**Published:** 2022-06-01

**Authors:** André Berchtold

**Affiliations:** Institute of Social Sciences & Centre LIVES, University of Lausanne, CH-1015 Lausanne, Switzerland; andre.berchtold@unil.ch

**Keywords:** COVID-19 pandemic, partial lockdown, well-being, adolescent-parent relationship

## Abstract

This study is based on two waves of data collected by the Swiss Household panel, the first one in 2019, before the beginning of the COVID-19 pandemic, and the second one in May–June 2020, just after the end of the partial lockdown that was decided by the Swiss government. We considered “couples” of adolescents (age 14–24, mean = 18.82, 51.96% female) and their parents living together (*n* = 431). Our main goal was to determine whether the evolution of the well-being among adolescents was similar to the evolution of the well-being among parents. Ten indicators of well-being were measured identically in both waves and for both adolescents and their parents. Results indicate that while almost all indicators of well-being decreased during partial lockdown for both adolescents and their parents, adolescents were more strongly impacted than their parents. Furthermore, the change observed in adolescents was virtually unaffected by the change observed in their parents, and vice versa. This research is a reminder that while different population groups may be affected differently by a sudden and extreme event, it is not only older people who will be most affected. Here, adolescents appear to have been more adversely affected than adults.

## 1. Introduction

The COVID-19 pandemic is a natural experiment that allows us to study in real conditions the effect of sudden and large-scale upheavals on human society [1]. In this article, we consider the relationship between adolescents and their parents and focus on the evolution of their respective well-being between before and after the partial lockdown decreed by the Swiss authorities. However, even though this research focuses on a specific event, it must be understood in a much broader sense, i.e., what are the consequences on adolescents of catastrophic events and are these consequences similar or not to those observed in older people [2]. The answer to this question may have very broad implications. Indeed, in the context of population aging that dominates the West, there is increasing pressure on younger generations to maintain intergenerational social solidarity, particularly through the financing of pension systems [3]. The realization that young people are not doing as well as their elders and that they are moreover greatly impacted by catastrophic events is therefore extremely worrying for the future of our society as a whole. This means paying special attention to the needs of adolescents in times of crisis. Thus, any investment in adolescents can only prove beneficial for the future [4].

Individual well-being also depends on collective well-being, i.e., the specific situation of one person can have an influence on those around him or her, and vice versa [5,6]. In the family domain in particular, recent research has shown that the well-being of the family as a whole takes precedence over the individual well-being of its members [7]. Furthermore, it has been shown that the level of psychological health of parents is positively correlated with the physical and psychological well-being of adolescents [8]. The question then arises as to what extent the same indicators of well-being measured in adolescents and their parents can not only be correlated but be at the same level. Furthermore, it is also of interest to determine to what extent the relationship between the well-being of adolescents and their parents is stable over time or whether it is likely to change rapidly, particularly if an unforeseen or catastrophic event occurs that is external and independent of the family situation.

Beyond the fact that they live together and thus jointly experience a whole series of facts of life, the relationship between adolescents and their parents is especially interesting because we are dealing with a comparison between people who are supposed to have achieved a certain stability in their life trajectory and young people who are, on the contrary, in a critical phase of their trajectory that should lead them from a situation of dependence on their parents to an adult situation of autonomy [9]. In this context, a sudden and important event could potentially exacerbate possible tensions within family relationships [10] or, on the contrary, serve as a point of diversion on which to transfer these tensions.

The COVID-19 pandemic is an excellent example of a situation in which a very large number of people were subjected to a sudden extraordinary event. In Switzerland, a partial lockdown was decided by the authorities between 8 March and 10 May 2020, after which the confinement measures were gradually relaxed. It can be noted that in international comparison, the level of restrictions that have been put in place by the Swiss government cannot be considered extreme. Indeed, according to the composite index calculated by the Government Response Tracker (https://www.bsg.ox.ac.uk/research/research-projects/COVID-19-government-response-tracker (accessed on 9 April 2020)) in the middle of the partial lockdown period, Switzerland had a score of 73.15, comparable to that of Germany, but much lower than that of the other neighboring countries: Austria (81.48), France (87.96) and Italy (91.67). Travel restrictions in Switzerland were not very significant and public transport was always available. However, due to the closure of all schools and the obligation to telework for the vast majority of working people, almost all adolescents in Switzerland spent the partial lockdown period living 24 h a day with their parents. This situation therefore represented an excellent opportunity to study the extent to which the evolution of a set of criteria related to the well-being and social life of adolescents is associated with the evolution of these same criteria in parents when all these people live together in an essentially closed environment.

It is known that many factors can positively or negatively influence the mood and well-being of young people [11]. However, most existing studies have been unable to consider this in the context of an event as important as the COVID-19 pandemic. Although exceptions exist [12], only young adults have been studied. What is unique about our study is that we were able to relate changes in well-being among both adolescents and their parents. To do this, we compared data measured in 2019, before the start of the COVID-19 pandemic, with other data collected in May and June 2020, just after the end of the partial lockdown. We hypothesized that: (1) changes in well-being measures among adolescents were related to those observed among their parents; but (2) greater changes have occurred among adolescents than among their parents. The first hypothesis stems from the idea that family climate as a whole is an important component of well-being [13,14], and the second hypothesis is related to the idea that, as a developing subpopulation, adolescents may be more affected by a sudden event that fundamentally alters their life pattern than adults [15,16].

## 2. Materials and Methods

This study is based on two waves of data collected by the Swiss Household panel [17], the first one in 2019, before the beginning of the COVID-19 pandemic, and the second one in May–June 2020, just after the end of the partial lockdown that was decided by the Swiss government [18]. We considered *n* = 431 “couples” of adolescents and their parents living together. Variables of interest were all measured identically in both waves and for both adolescents and their parents. They cover several indicators that can be related to the general concept of well-being, including somatic and psychological health (overall health level, physical activity, satisfaction with life in general, feeling alone, negative feelings, energy/optimism, and perceived stress), relationships (satisfaction with relationships, trust in other people), and satisfaction with leisure. Most of these measures are standard tools from the literature: The satisfaction with life measure is based on Diener et al. [19]; the satisfaction with leisure and satisfaction with relationships measures are based on Diener at al. [20]; the energy/optimism and negative feelings measures are based on Watson et al. [21]; the trust in other people measure is based on Rosenberg [22]; the feeling alone and perceived stress measures were developed by the Swiss Household Panel (https://forscenter.ch/projects/swiss-household-panel/ (accessed on 20 May 2022)); the physical activity and overall health measures were adapted from the Swiss Health Survey (https://www.bfs.admin.ch/bfs/fr/home/statistiques/sante/enquetes/sgb.html (accessed on 20 May 2022)). Most variables were measured on a scale from zero (not at all satisfied) to ten (completely satisfied), except for the overall health (five-point scale), physical activity (number of days per week with a physical activity of at least 30 min), and perceived stress (five-point scale). In all cases, a higher value of the indicator indicates a better situation for the person. To achieve this, three of the indicators had their measurement scales reversed. These are negative feelings, feeling alone, and perceived stress. It may seem strange that a higher level on the stress scale, for example, actually corresponds to less stress, but this makes sense for the interpretation, because in this research we are primarily interested in the evolution of the level between two distinct moments. Therefore, a negative change on the stress scale corresponds to an increase in stress, while a positive change corresponds to an improvement. Control variables include age in years, gender (female, male), being born in Switzerland (yes, no), interview language (French, German, Italian), living in a single-parent family (yes, no), and currently in education (yes, no, for adolescents only).

For some adolescents, data from one of the two parents was not available, either because they lived in a single-parent family or because one of the parents did not answer at least one of the questionnaires. To solve this issue, we adopted the following strategy: Each calculation involving the parents’ data was replicated 100 times. When data from only one of the two parents were available, they were used for each replication. In contrast, when data from both parents were available, a random selection of one of the two parents was made separately for each of the 100 replications. To ensure that the results were consistent across calculations, the same selection was used for each calculation. The final results are the aggregated results from the 100 replications.

First, the main characteristics of the sample of adolescents were described (numbers and percentages for categorical variables, range, mean, and standard deviation for continuous variables). Then, the average score of each of the 10 indicators analyzed was calculated before and after the partial lockdown, and the average evolution was deduced. A Student’s *t*-test was performed to determine whether the change was significant or not. The same calculations were performed for the parents, and then a paired Student’s *t*-test was performed between changes experienced by adolescents and their parents. Finally, linear regression models were calculated to explain the evolution of each indicator observed among adolescents as a function of the same indicator observed among the parents and control variables (gender, age, being born in Switzerland, language of the interview, and living in a single-parent family). The same models were also calculated to explain the evolution of each indicator among parents as a function of the evolution of the same indicator among adolescents as well as control variables, the latter being measured among the parents.

To ensure the best possible representativeness of the sample, all calculations were made using the weights provided by the Swiss Household Panel. This explains why most of the numbers reported in the Results Section are not integer values. The Benjamini–Hochberg method [23] was used to correct for *p*-values in the case of multiple comparisons on the 10 well-being indicators. All calculations were performed using the open source statistical software R [24]. The Type I error was set to 5%.

## 3. Results

The sample comprises *n* = 431 adolescents (51.96% female, mean age 18.82 years) and Table 1 describes its main characteristics. The weights used for all calculations make this sample representative of the population structure of the same age group living in Switzerland (https://www.bfs.admin.ch/bfs/fr/home.html (accessed on 8 March 2022)).

Table 2 and Figure 1 describe the evolution of the 10 well-being indicators during the partial lockdown that was set up in Switzerland. In Figure 1, the mean scores of adolescents are on the *x*-axis and those of parents on the *y*-axis, with the 45-degree dashed line visualizing the equality between the two scores. For each of the 10 indicators, the arrow represents the change from the pre-lockdown situation (beginning of the arrow) to the post-lockdown situation (tip of the arrow). Indicators that are better in adolescents both before and after lockdown are in green, those that are better in parents both before and after lockdown are in blue, and those that were better in adolescents before lockdown but became better in parents after lockdown are in red. While half of the indicators measured in adolescents were better than those of their parents before the partial lockdown, this is no longer the case after the partial lockdown and the parents obtained better scores on 8 of the 10 indicators, the two exceptions being the satisfaction with leisure activities and the overall health. As indicated in Table 2, however, most of the differences were not significant. Moreover, even if they differed, the scores of adolescents and parents stayed always in a comparable range.

Globally, all indicators worsened during the partial lockdown period, the only two exceptions being the level of perceived stress and the overall health of both adolescents and their parents. All changes between the pre- and post-lockdown situations were statistically significant, but if we now compare the level of change observed in the adolescents with that of their parents, we can see that this level was statistically different for only two indicators: satisfaction with life, and satisfaction in relationships, with in both cases a greater deterioration in the adolescents. For the other eight indicators, the level of change observed in adolescents did not differ statistically from that of parents.

In order to refine these results, Table 3 summarizes the regression models calculated to explain each adolescent well-being indicator as a function of the same parent indicator and covariates (left-hand side of the table), and each parent well-being indicator as a function of the same adolescent indicator and covariates (right-hand side of the table). With the exception of the changes in the parents’ energy/optimism level and physical activity level, which can be partly explained by the same indicators measured in the adolescents, no other changes can be explained by the change in the same indicator in the other family member. On the other hand, some of the covariates used are significant, mainly in the explanatory models for the adolescent indicators. Overall, girls experienced worse changes in their well-being indicators during partial lockdown than boys, and younger adolescents experienced more negative changes than older ones. Regarding the satisfaction with relationships, German-speaking adolescents had a better evolution than French-speaking adolescents, and the evolution was worse in single-parent families.

## 4. Discussion

The aim of this research was to compare the evolution of adolescents and their parents in Switzerland during the partial lockdown due to the COVID-19 pandemic through 10 indicators representing different facets of well-being. Our hypotheses were (1) that there would be a link between the evolution of the indicators of adolescents and parents; (2) that the evolution would be stronger for adolescents than for parents. However, our analyses lead us to put these two hypotheses into perspective. Indeed, although Table 2 and Figure 1 indicate that the indicators always evolved in the same direction on average (increase or decrease) for adolescents and their parents, the evolution observed for individuals in one of the two groups can almost never be explained by the evolution observed in the other group (Table 3). Moreover, the magnitude of the change observed among adolescents is only significantly greater than that of their parents in 2 indicators out of 10 (satisfaction with life and satisfaction with relationships). This leads us to hypothesize that the evolution of well-being may have been more influenced by factors external to the family and induced by the general pandemic situation, which is compatible with the significant drop in the average level of almost all the indicators, and not by the evolution of this well-being in specific individuals, parents or adolescents, since the evolution of an indicator in one or the other person hardly explains its evolution in the other person. In this sense, our results are in line with the literature according to which the study of well-being implies a relationship between individual and collective levels [25].

It goes without saying that the exceptional situation created by the COVID-19 pandemic has had a significant impact on families and the way they live together [26]. It is even possible to hypothesize that the forced limitation of youth mobility could have effects on their longer-term mobility, particularly in relation to the usual flows of youth leaving or returning to the family home [27]. Although many aspects of life have already been studied in the context of COVID-19, such as the impact on education [28] or the family’s financial situation [29], the impact on the relationship between adolescents and parents in terms of well-being had not yet been examined. Hence, this research was necessary in order to provide a better understanding of the evolution of well-being in adolescents and their parents following an exceptional event. It appears that the partial lockdown, and perhaps more generally the COVID-19 pandemic, had a greater impact on adolescents than on their parents, but with variations among adolescents, with girls and younger adolescents being more affected. This is reflected in the fact that while before the partial lockdown half of the indicators were better for adolescents than for their parents, only two were better at the end of the partial lockdown. Conversely, the indicators that were better for parents than for adolescents before partial lockdown remained so after it. This can be related to research showing that younger individuals are more influenced by other individuals than older people [15]. By extension, it can be assumed that this same phenomenon also occurs when the influence is not due to individuals but rather to a situation, in this case the COVID-19 pandemic and partial lockdown. Another possible explanation is based on the resource accumulation theory [30]. Developed in the context of life-course analysis, this theory states that throughout his or her life, an individual can accumulate different types of resources that he or she can then mobilize to deal with difficult situations. Following this theory, older adults should have more resources on average than adolescents, which explains why they may be relatively less affected by a negative event.

It may seem surprising at first sight that during the partial lockdown period, on average, the general health of both adolescents and parents included in this research improved. However, this confirms some known data, namely that during this period, various infectious diseases such as tuberculosis and legionellosis declined, due to the limitation of interpersonal contacts and therefore, the risks of contamination (https://www.bfs.admin.ch/bfs/fr/home.html (accessed on 12 March 2022)) [31,32]. In addition, the widespread use of masks, hydro-alcoholic gels, and social distancing certainly also had an impact beyond the prevention of COVID-19 alone. The same reduction was also observed for other diseases such as malaria (effect due to the sharp drop in travel), syphilis (limitation of the number of occasional sexual partners and prostitution), or more simply, the common cold. As a result, despite the COVID-19 epidemic, some people paradoxically felt healthier than in ordinary times.

The main strength of this research is that it is based on data collected as close as possible to the impacting event, that is during the COVID-19 pandemic, just before and after the Swiss partial lockdown. Moreover, these data are extremely original since they allow the evolution of the well-being of adolescents and their parents to be linked. However, several limitations must also be mentioned. First, the sample size is relatively small for a quantitative study, although it is similar to other quantitative studies conducted during the COVID-19 pandemic [12]. Secondly, even in the case of nuclear families, data from both parents were not always available. One possibility would have been to fully impute the data of the missing parent, but this would not have made the comparison with single-parent families any easier. For this reason, we opted for a bootstrap approach by randomly selecting which of the two parents would be used in each replication of the calculations. Furthermore, some variables had missing data and it would have been possible to impute them so that all calculations could be performed on the whole sample. However, due to the very personal nature of the variables analyzed (well-being indicators), we could not find satisfactory imputation models. As a result, we preferred not to impute missing data. Finally, given the lack of significant links between the adolescents’ indicators and those of their parents, synthetic indicators were constructed by adding up several of the available indicators. However, none of these synthetic indicators reached the minimum threshold of a Cronbach’s alpha of 0.7 and were therefore dropped. It is also possible to criticize the choice of indicators we used to represent well-being, as they do not conform to the various scales developed in the literature [33], but none of these scales were available in the data at our disposal. More generally, it should also be pointed out that the data collected at the end of the partial lockdown by the Swiss Household Panel are much more limited than those of the usual annual waves and thus we did not have contextual data on family climate that could have been used particularly in the framework of a multilevel modeling.

## 5. Conclusions

This research shows that the well-being of adolescents and their parents under the 2020 partial lockdown regime in Switzerland declined overall, with adolescents generally being more severely affected. However, it is difficult to generalize these results to other countries, as the modalities of the lockdowns, both in terms of duration and level of restrictions, have varied widely across countries. Therefore, it would be very interesting to replicate this same research in other countries, but we are not aware of any data collected under the same conditions as the Swiss household panel that would allow this to be carried out. From a methodological point of view, this shows the importance of being able to work quickly and to have representative panels that can be mobilized under all circumstances. For this reason, we advise government agencies involved in funding scientific research to set up data collection and analysis infrastructures that allow them to react quickly when natural study opportunities such as that brought about by the COVID-19 pandemic arise. It is obviously not a question of wishing for such situations to occur, but of being able to exploit them to the best of our ability in order to develop our knowledge, and thus perhaps avoid new catastrophic events. On a more societal level, this research shows that certain population groups may be more vulnerable to sudden changes and that these groups are not only to be found among the oldest people, but also among young people, adolescents in this case. As a take-home message, we want to emphasize that even if the population as a whole tends to age, at least in Western countries, it is essential not to forget that it is the younger generation that will allow the social fabric to be sustained through the mechanisms of intergenerational solidarity. It is therefore essential to give special importance to the development of adolescents and young adults and to allow them to flourish, whatever the circumstances.

## Figures and Tables

**Figure 1 ijerph-19-06789-f001:**
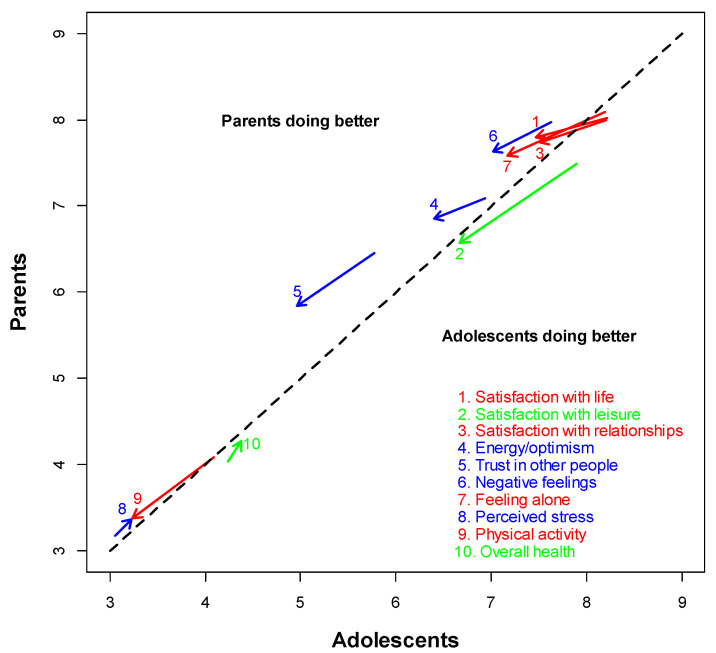
Evolution of the well-being of adolescents and parents during the partial lockdown.

**Table 1 ijerph-19-06789-t001:** Description of the sample of adolescents.

Variable	Frequency	Percentage
Gender		
Female	223.90	51.96
Male	207.10	48.04
Age: min–max, mean (sd)	14–24, 18.82 (3.61)	
Born in Switzerland		
No	87.51	20.30
Yes	343.49	79.70
Language		
French	136.39	31.64
German	249.65	57.92
Italian	44.96	10.43
Single-parent family		
No	358.99	83.29
Yes	72.01	16.71
Among single-parent family:		
living with their mother only	60.96	84.66
living with their father only	11.04	15.34
In education		
No	74.43	17.27
Yes	356.57	82.73
Age of the mother (*n* = 419.95)	35–65, 50.46 (5.95)	
Age of the father (*n* = 370.03)	38–72, 53.72 (6.99)	

**Table 2 ijerph-19-06789-t002:** Evolution of well-being during the partial lockdown. We provide the number of usable observations, the average value of each indicator before and after the partial lockdown, and the corresponding average evolution separately for adolescents and their parents. Evolutions significant at the 95% level according to a Benjamini–Hochberg adjusted *t*-test appear in bold. Similarly, when the “before” scores are significantly different between adolescents and parents they appear in bold, and the same is true for the “after” scores. The last column provides the *p*-value of a paired *t*-test between the evolution of the adolescents and their parents.

	Adolescents				Parents				
Indicators	*n*	Before	After	Evolution	*n*	Before	After	Evolution	*p*-Value
Satisfaction with life	431	8.21	7.47	**−0.74**	428.03	8.02	7.80	**−0.22**	**0.020**
Satisfaction with leisure	427	**7.90**	6.67	**−1.23**	427.20	**7.49**	6.57	**−0.95**	0.212
Satisfaction with relationships	428	8.20	7.51	**−0.68**	428.00	7.99	7.74	**−0.26**	**0.045**
Energy/optimism	431	6.94	**6.40**	**−0.53**	426.50	7.09	**6.85**	**−0.25**	0.193
Trust in other people	431	**5.77**	**4.96**	**−0.82**	425.29	**6.45**	**5.84**	**−0.60**	0.330
Negative feelings	424	7.63	**7.02**	**−0.60**	427.08	7.97	**7.63**	**−0.35**	0.232
Feeling alone	429	8.19	7.17	**−1.01**	430.00	8.09	7.58	**−0.51**	0.160
Perceived stress	431	3.06	3.23	**0.17**	429.46	3.18	3.37	**0.20**	0.754
Physical activity (days)	320	4.09	3.24	**−0.85**	351.25	4.09	3.38	**−0.72**	0.609
Overall health	431	**4.24**	4.38	**0.14**	431.00	**4.04**	4.27	**0.24**	0.263

**Table 3 ijerph-19-06789-t003:** Regression models for the explanation of the evolution of an indicator of the adolescents in function of the same indicator among their parents, including control variables. The right part of the table provides the same computation for the indicators of the parents. For each significant covariate, we provide the coefficient and the *p*-value.

		Adolescents in Function of Parents			Parents in Function of Adolescents		
Indicators	*n*	Coef	*p*	Significant Covariates	Coef	*p*	Significant Covariates
Satisfaction with life	423.03	−0.08	0.371	Gender (−0.52, 0.026)	−0.05	0.313	-
Satisfaction with leisure	423.20	0.05	0.490	-	0.04	0.443	Gender (−0.79, 0.004)
Satisfaction with relationships	418.00	0.06	0.372	Language GvsF (1.07, <0.001) Single-parent (−0.73, 0.018)	0.05	0.257	-
Energy/optimism	421.50	0.15	0.059	Age (0.10, 0.010)	0.10	0.050	-
Trust in other people	413.78	0.05	0.495	Gender (−0.53, 0.049) Age (0.11, 0.013)	0.03	0.511	-
Negative feelings	420.08	0.10	0.187	-	0.07	0.197	-
Feeling alone	423.00	−0.03	0.491	-	−0.05	0.423	-
Perceived stress	428.46	−0.03	0.644	Gender (−0.32, 0.025)	−0.04	0.584	-
Physical activity (days)	270.14	0.15	0.064	Gender (−0.76, 0.021)	0.20	0.049	-
Overall health	430.00	0.05	0.447	Age (0.03, 0.043)	0.04	0.467	-

## Data Availability

The data used in this article can be obtained from the Swiss Household Panel: https://forscenter.ch/projects/swiss-household-panel/ (accessed on 20 May 2022).

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
