# Peer review of "Relationship between the COVID-19 Pandemic and the Well-Being of Adolescents and Their Parents in Switzerland"

_ijerph, 2022, doi:10.3390/ijerph19116789_

Round 1
Reviewer 1 Report
This is an interesting study. However, there are still some shortcomings in the research design and theoretical analysis. A few comments for your reference:
(1) The introduction needs rewriting. The present introduction only describes the whole research in the context of COVID-19, but does not touch on the core of the research, namely the importance of the research, the key scientific issues to be solved and the marginal contribution of the research. Why is it important that the impact of COVID-19 is different for different groups (parents versus children)? In fact, it is an objective fact that there are correlations and differences in welfare among different groups regardless of whether COVID-19 is present or not, and it is not necessary to do research to realize this conclusion.
(2) The marginal contribution of research is not clear. At the same time, the research also lacks rigorous theoretical analysis, and the research hypothesis is particularly "hasty" without any theoretical support. In addition, the subsequent empirical studies were more about comparing the welfare of the two groups before and after the policy impact of COVID-19 without regression analysis, so the author's hypothesis 1 could not be well verified in fact.
(3) The conclusion and discussion are too simple.
Author Response
Reviewer 1
This is an interesting study. However, there are still some shortcomings in the research design and theoretical analysis. A few comments for your reference:
(1) The introduction needs rewriting. The present introduction only describes the whole research in the context of COVID-19, but does not touch on the core of the research, namely the importance of the research, the key scientific issues to be solved and the marginal contribution of the research. Why is it important that the impact of COVID-19 is different for different groups (parents versus children)? In fact, it is an objective fact that there are correlations and differences in welfare among different groups regardless of whether COVID-19 is present or not, and it is not necessary to do research to realize this conclusion.
Thank you for the comment, which we fully understand. It is true that only the specific context of the research was described in the first version of the article and that more general information about its scope was missing. In particular, it was not clear how a different evolution of adolescents compared to older people like their parents could be especially important. Therefore, the article now begins with a brand new paragraph:
“The Covid-19 pandemic is a natural experiment that allows us to study in real conditions the effect of sudden and large-scale upheavals on human society [1]. In this article, we consider the relationship between adolescents and their parents and focus on the evolution of their respective well-being between before and after the partial lockdown decreed by the Swiss authorities. However, even though this research focus-es on a specific event, it must be understood in a much broader sense, i.e., what are the consequences on adolescents of catastrophic events and are these consequences similar or not to those observed in older people [2]. The answer to this question may have very broad implications. Indeed, in the context of population aging that dominates the West, there is increasing pressure on younger generations to maintain intergeneration-al social solidarity, particularly through the financing of pension systems [3]. The real-ization that young people are not doing as well as their elders and that they are more-over greatly impacted by catastrophic events is therefore extremely worrying for the future of our society as a whole. This means paying special attention to the needs of adolescents in times of crisis. Thus, any investment in adolescents can only prove beneficial for the future [4].”
In addition, the transition to the hypotheses has been improved and our hypotheses are now supported by references from the literature.
(2) The marginal contribution of research is not clear. At the same time, the research also lacks rigorous theoretical analysis, and the research hypothesis is particularly "hasty" without any theoretical support. In addition, the subsequent empirical studies were more about comparing the welfare of the two groups before and after the policy impact of COVID-19 without regression analysis, so the author's hypothesis 1 could not be well verified in fact.
We agree that the introduction to the article may have been too focused on the Covid-19 pandemic issue alone, and as such the research question may have been unclear and the hypotheses lacked support. We hope that the changes to the introduction are along the lines you were hoping for (see the response to your previous comment). In particular, the last paragraph of the introduction now begins like this:
“It is known that many factors can positively or negatively influence the mood and well-being of young people [11]. However, most existing studies have been unable to consider this in the context of an event as important as the covid-19 pandemic. Although exceptions exist [12], only young adults have been studied. What is unique about our study is that we were able to relate changes in well-being among both adolescents and their parents.”
On the other hand, we would like to point out that Table 3 of the article does report the results of linear regressions comparing the evolution of indicators of well-being of adolescents and their parents. The left-hand side of the table shows the evolution before/after confinement of adolescents according to the evolution of the same indicator measured in their parents, and the right-hand side of the table shows the opposite. It seems to us that we have analyses that support our hypotheses.
(3) The conclusion and discussion are too simple.
The discussion and conclusion of the article have been improved, including the addition of more specific references to the existing literature and to certain concepts. For example:
“It goes without saying that the exceptional situation created by the covid-19 pandemic has had a significant impact on families and the way they live together [26]. It is even possible to hypothesize that the forced limitation of youth mobility could have effects on their longer-term mobility, particularly in relation to the usual flows of youth leaving or returning to the family home [27]. Although many aspects of life have already been studied in the context of covid-19, such as the impact on education [28] or the family's financial situation [29], the impact on the relationship between adolescents and parents in terms of well-being had not yet been examined.”
or
“As a take-home message, we want to emphasize that even if the population as a whole tends to age, at least in Western countries, it is essential not to forget that it is the younger generation that will allow the social fabric to be sustained through the mechanisms of intergenerational solidarity. It is therefore essential to give special importance to the development of adolescents and young adults and to allow them to flourish, whatever the circumstances.”
Reviewer 2 Report
The manuscrip entitled "Impact of the Covid-19 pandemic on the well-being of adolescents and their parents in Switzerland" regards the impact of pandemic on well-being in adolescent and their parents.
Some changes and integrations are necessary
Please add a take home message at the end of the abstract
Please see the work of Lopez and colleagues (DOI: 10.1111/aphw.12268), and Germani (doi.org/10.1111/sjop.12616; doi.org/10.3390/ijerph17103497) in order to improve the introduction. The construct of well-being must be specified.
At the end of the introduction, please clarify the aim of the study
Please specify all the measures used, which questionnaires were used and their reliability.
Please bettere explain the strategy used to obtain 100 replication
Please describe the participants and add the sample size in the method
In the discussion, please read the results in the light of the existing literature. Moreover add the limination of the study and better specify the future directions
Please check some typos throughout the manuscript
Author Response
Reviewer 2
The manuscrip entitled "Impact of the Covid-19 pandemic on the well-being of adolescents and their parents in Switzerland" regards the impact of pandemic on well-being in adolescent and their parents.
Some changes and integrations are necessary
Please add a take home message at the end of the abstract
Two messages were added in the conclusion, the first of a methodological nature and the second more focused on the well-being of younger generations:
“For this reason, we advise government agencies involved in funding scientific research to set up data collection and analysis infrastructures that allow them to react quickly when natural study opportunities such as that brought about by the covid-19 pandemic arise. It is obviously not a question of wishing for such situations to occur, but of being able to exploit them to the best of our ability in order to develop our knowledge, and thus perhaps to avoid new catastrophic events.”
and
“As a take-home message, we want to emphasize that even if the population as a whole tends to age, at least in Western countries, it is essential not to forget that it is the younger generation that will allow the social fabric to be sustained through the mechanisms of intergenerational solidarity. It is therefore essential to give special importance to the development of adolescents and young adults and to allow them to flourish, whatever the circumstances.”
Please see the work of Lopez and colleagues (DOI: 10.1111/aphw.12268), and Germani (doi.org/10.1111/sjop.12616; doi.org/10.3390/ijerph17103497) in order to improve the introduction. The construct of well-being must be specified.
The introduction has been completely redesigned to highlight the existing literature. In particular, a new introductory paragraph explains the general scope of this article beyond the specific context of the covid-19 pandemic:
“The Covid-19 pandemic is a natural experiment that allows us to study in real conditions the effect of sudden and large-scale upheavals on human society [1]. In this article, we consider the relationship between adolescents and their parents and focus on the evolution of their respective well-being between before and after the partial lockdown decreed by the Swiss authorities. However, even though this research focus-es on a specific event, it must be understood in a much broader sense, i.e., what are the consequences on adolescents of catastrophic events and are these consequences similar or not to those observed in older people [2]. The answer to this question may have very broad implications. Indeed, in the context of population aging that dominates the West, there is increasing pressure on younger generations to maintain intergeneration-al social solidarity, particularly through the financing of pension systems [3]. The real-ization that young people are not doing as well as their elders and that they are more-over greatly impacted by catastrophic events is therefore extremely worrying for the future of our society as a whole. This mean paying special attention to the needs of adolescents in times of crisis. Thus, any investment in adolescents can only prove beneficial for the future [4].”
In addition, many references have been added and the transition to the assumptions has been redesigned:
“It is known that many factors can positively or negatively influence the mood and well-being of young people [11]. However, most existing studies have been unable to consider this in the context of an event as important as the covid-19 pandemic. Although exceptions exist [12], only young adults have been studied. What is unique about our study is that we were able to relate changes in well-being among both adolescents and their parents.”
As we specify in the article, the Swiss Household Panel data used in this study were not intended to study well-being and therefore they did not include any existing reliable measure. We did try to construct a synthetic indicator from the available measures, but its internal consistency proved to be much too low (Cronbach's alpha below 0.7). This is why, from a methodological point of view, it seemed preferable to consider the 10 measures separately.
At the end of the introduction, please clarify the aim of the study
We hope that the various changes in the introduction will provide a better understanding of the value of studying the relationship between the well-being of adolescents and that of older people, in this case their parents.
Please specify all the measures used, which questionnaires were used and their reliability.
The “Satisfaction with life” measure is based on Diener, Ed, Robert A. Emmons, Randy J. Larsen, and Sharon Griffin. 1985. "The satisfaction with life scale." Journal of personality assessment 49(1):71-75. The “Satisfaction with leisure” and “Satisfaction with relationships measures are based on Diener, Ed, Mark Eunkook Suh, Richard E. Lucas, and Heidi L. Smith. 1999. "Subjective well-being: Three decades of progress." Psychological Bulletin 125(2):276-302. The “Energy/optimism” and “Negative feelings” measures are based on Watson, David, Lee A. Clark, and Auke Tellegen. 1988. "Development and validation of brief measures of positive and negative affect: The PANAS scales." Journal of Personality and Social Psychology 54(6):1063-70. The “Trust in other people” measure is based on Rosenberg, M. (1956). “Misanthropy and political ideology”. American Sociological Review, 21(6), 690–695. The “Feeling alone” and “Perceived stress” measures were developed by the Swiss Household Panel. The “Physical activity” and “Overall health” measures were adapted from the Swiss Health Survey. https://www.bfs.admin.ch/bfs/fr/home/statistiques/sante/enquetes/sgb.html (accessed May 18, 2022). More information is available on the website of the Swiss Household Panel: https://forscenter.ch/projects/swiss-household-panel/ (accessed May 18, 2022).
All references are now included in the article.
Please better explain the strategy used to obtain 100 replications.
All calculations involving the parents were replicated 100 times. When data from only one of the two parents were available, they were used for each replication. In contrast, when data from both parents were available, a random selection of one of the two parents was made separately for each of the 100 replications. To ensure that the results were consistent across calculations, the same selection was used for each calculation. This procedure is now better explained in the methods section of the article.
Please describe the participants and add the sample size in the method
The sample size was already shown in the results, but is now also given in the Methods section. The main characteristics of the adolescents in the sample appear in Table 1 and we have completed this table by adding the main characteristics of the parents.
In the discussion, please read the results in the light of the existing literature. Moreover add the limitation of the study and better specify the future directions
Clarifications were made to the limitations of the study (last paragraph of the discussion). In addition, the discussion has been improved by introducing new references and concepts from the literature. In addition, the conclusion has been modified to include both methodological and societal recommendations.
Please check some typos throughout the manuscript.
Thank you. The manuscript has been fully proofread to eliminate typos.
Round 2
Reviewer 1 Report
Thank you for your reply and revision. However, the title of the paper studies causality, while in the text the author does correlation, and correlation cannot replace causality. Therefore, what the author wanted to achieve was not accomplished.
Author Response
I think your comment is related to the use of the word "Impact" in the title. As you rightly say, this article does not purport to try to establish causality between the Covid-19 pandemic and partial lockdown on the one hand, and changes in the well-being of adolescents and their parents on the other. The data necessary for this are not available, in particular to be able to rule out all other potentially competing causes. This is why the article speaks of correlation and not causatity. In order to reflect this more accurately, the title has been changed to "Relationship between the Covid-19 pandemic and the well-being of adolescents and their parents in Switzerland".
Reviewer 2 Report
The authors replied all the questions raised. The manuscript can be accepted in the present form.
Author Response
Thank you very much.